# Tumor copy number alteration burden is a pan-cancer prognostic factor associated with recurrence and death

Haley Hieronymus[1], Rajmohan Murali[2], Amy Tin[3], Kamlesh Yadav[4], Wassim Abida[1,5], Henrik Moller[6], Daniel Berney[7], Howard Scher[5,8], Brett Carver[9], Peter Scardino[9], Nikolaus Schultz[10], Barry Taylor[1,3,10], Andrew Vickers[3], Jack Cuzick[11], Charles L Sawyers[1,12]*

[1]Human Oncology and Pathogenesis Program, Memorial Sloan Kettering Cancer Center, New York, United States; [2]Department of Pathology, Memorial Sloan Kettering Cancer Center, New York, United States; [3]Department of Epidemiology and Biostatistics, Memorial Sloan Kettering Cancer Center, New York, United States; [4]Department of Urology, Icahn School of Medicine at Mount Sinai, New York, United States; [5]Genitourinary Oncology Service, Department of Medicine, Memorial Sloan Kettering Cancer Center, New York, United States; [6]Department of Cancer Epidemiology, Population and Global Health, King's College London, London, United Kingdom; [7]Department of Molecular Oncology, Barts Cancer Institute, Queen Mary University of London, London, United Kingdom; [8]Department of Medicine, Weill Cornell Medical College, New York, United States; [9]Department of Urology, Memorial Sloan Kettering Cancer Center, New York, United States; [10]Marie-Josée and Henry R. Kravis Center for Molecular Oncology, Memorial Sloan Kettering Cancer Center, New York, United States; [11]Centre for Cancer Prevention, Wolfson Institute of Preventive Medicine, Queen Mary University of London, London, United Kingdom; [12]Howard Hughes Medical Institute, Chevy Chase, United States

*For correspondence:
sawyersc@mskcc.org

**Abstract** The level of copy number alteration (CNA), termed CNA burden, in the tumor genome is associated with recurrence of primary prostate cancer. Whether CNA burden is associated with prostate cancer survival or outcomes in other cancers is unknown. We analyzed the CNA landscape of conservatively treated prostate cancer in a biopsy and transurethral resection cohort, reflecting an increasingly common treatment approach. We find that CNA burden is prognostic for cancer-specific death, independent of standard clinical prognosticators. More broadly, we find CNA burden is significantly associated with disease-free and overall survival in primary breast, endometrial, renal clear cell, thyroid, and colorectal cancer in TCGA cohorts. To assess clinical applicability, we validated these findings in an independent pan-cancer cohort of patients whose tumors were sequenced using a clinically-certified next generation sequencing assay (MSK-IMPACT), where prognostic value varied based on cancer type. This prognostic association was affected by incorporating tumor purity in some cohorts. Overall, CNA burden of primary and metastatic tumors is a prognostic factor, potentially modulated by sample purity and measurable by current clinical sequencing.
DOI: https://doi.org/10.7554/eLife.37294.001

**eLife digest** Cancer cells carry different types of mutations that are associated with the cell starting to multiply uncontrollably. Certain changes only affect one or a few letters of the genetic code. Others, known as copy number alterations, or CNA, involve larger portions of the genome that can either be lost (deletions) or duplicated (amplifications). Tumors in different patients carry variable amounts of these deletions or amplifications, which together are known as the CNA burden.

New technologies allow scientists to scan the genomes of tumors and examine the type of mutations present in each patient. The results can help to decide on the best course of action. For example, in prostate cancer, patients whose tumors have a high CNA burden are at greater risk of relapse after treatment. However, it has been unclear whether these people also have lower survival rates, and if CNA burden can predict outcome of other types of cancers.

Hieronymus et al. conducted genetic analyses on over a hundred samples from prostate cancer patients who were not treated with surgery or radiation. The results showed that a higher CNA burden in the tumors is correlated with more deaths due to the disease. The findings in prostate cancer were also true across different types of cancers. These conclusions also emerged when Hieronymus et al. then looked at genomic data obtained from patients with various cancers using a different DNA sequencing test, which is certified for clinical use. This demonstrates that CNA burden could be a useful marker in clinical settings to help assess risk in cancer patients.
DOI: https://doi.org/10.7554/eLife.37294.002

## Introduction

Somatic copy number alterations (CNAs) are nearly ubiquitous in cancer (*Zack et al., 2013*; *Heitzer et al., 2016*) and alter a greater portion of the cancer genome than any other type of somatic genetic alteration (*Heitzer et al., 2016*). Different cancer types vary in their balance of copy number alterations to somatic point mutations, with prostate cancer having relatively high rates of CNA compared to point mutation. Given the prevalence of CNAs in cancer, significant effort has been directed towards identifying specific CNAs associated with cancer clinical characteristics and prognosis as well as the potential driver genes they contain (*Liang et al., 2016*; *Wang et al., 2016*; *Nibourel et al., 2017*). There are well demonstrated associations between specific CNAs and CNA signatures to cancer state and characteristics (*Visakorpi et al., 1995*; *Williams et al., 2014*; *Taylor et al., 2010*). CNV patterns or clusters have been associated with high Gleason prostate cancer (Gleason 8 + compared to Gleason 6–7 [*Williams et al., 2014*]) and recurrent disease (compared to primary [*Visakorpi et al., 1995*; *Cancer Genome Atlas Research Network, 2015*; *Viswanathan et al., 2018*]). Nonetheless, most CNAs are large, (*Zack et al., 2013*; *Beroukhim et al., 2010*) and their associations with cancer outcome may not be well identified by gene-specific approaches. Increasing evidence indicates that large CNAs harbor multiple drivers (*Tschaharganeh et al., 2016*; *Liu et al., 2016*), emphasizing the need to study their biological and clinical significance beyond individual gene-focused standpoints.

The CNA burden of a tumor is the degree to which a tumor's genome is altered as a percentage of genome length and represents a fundamental measure of genome copy number alteration level. As such, tumor CNA burden, rather than individual CNAs, may be associated with cancer outcomes. While tumor mutational burden (TMB) predicts response to immunotherapy across multiple cancer types (*Bergerot et al., 2018*; *Goodman et al., 2017*), tumor CNA burden may be prognostic for outcomes such as recurrence and survival. Indeed, we and others have previously found CNA burden and genome-wide CNA patterns to be associated with biochemical recurrence and metastasis in primary prostate cancer, the most common cancer in men, across multiple cohorts (*Taylor et al., 2010*; *Hieronymus et al., 2014*; *Camacho et al., 2017*). This prognostic significance of tumor CNA burden extends to low and intermediate risk prostate cancer (Gleason scores of 7 and less) (*Hieronymus et al., 2014*) and has the potential to better stratify risk in patients who are considering conservative treatment approaches such as active surveillance to reduce overtreatment (*Chen et al., 2016*; *Tosoian et al., 2016*).

In addition to questions about the prognostic potential and overall landscape of CNA in conservatively treated prostate cancer, it is unknown whether CNA burden is prognostic for prostate

cancer survival, rather than only recurrence and metastasis. Nor is it known whether the prognostic significance of tumor CNA burden extends to other cancer types. Here we set out to address these questions, as well as whether tumor CNA burden can be prognostic in a clinical practice setting, including (i) in cancers treated conservatively rather than through immediate surgery or radiation, (ii) in biopsy or resection samples, and (iii) using a clinical targeted sequencing that allows rapid and cost-effective measurement of tumor CNA burden.

To address these questions, we first examine the genomic CNA landscape of conservatively treated prostate cancer in more than a hundred diagnostic biopsy and resection specimens from a conservatively treated cohort; this cohort consisted of patients with localized prostate who were not treated with surgery or radiation within six months of diagnosis. We demonstrate that tumor CNA burden is associated with cancer-specific death, independent of standard clinical predictors. To explore the prognostic significance of tumor CNA burden more broadly in other cancer types, we find that tumor CNA burden is also associated with disease-free and overall survival in TCGA cohorts of primary breast, endometrial, renal clear cell, thyroid, and colorectal cancer in addition to prostate cancer, with the degree of association varying in some cancer types. We then establish the clinical feasibility of measuring tumor CNA burden using the FDA-cleared MSK-IMPACT clinical next generation sequencing (NGS) assay and confirm that tumor CNA burden is associated with overall and disease-specific survival in both primary and metastatic tumors across cancer types. In all, we demonstrate that tumor CNA burden is a prognostic factor associated with cancer recurrence and death in multiple cancer types, including in conservatively treated prostate cancer which would benefit from increased risk stratification.

## Results

### The genomic copy number landscape of conservatively treated prostate cancer

To explore the genomic copy number landscape of conservatively treated prostate cancer, we set out to analyze copy number alteration (CNA) in cancer obtained non-surgically through biopsy and transurethral prostate resection (TURP) using a widely studied, conservatively treated primary prostate cancer cohort (*Cuzick et al., 2006*). This retrospective Transatlantic Prostate Group 1 (TAPG1) cohort (n = 1675) consists of men below age 76 with clinically localized prostate cancer and prostate-specific antigen (PSA) below 100 ng/ml who did not receive surgery or radiation within 6 months of diagnosis (*Cuzick et al., 2006*). This population-based cohort was drawn from six cancer registries in Great Britain, and the majority of the cohort was followed without treatment, while a subset received hormonal therapy. The original diagnostic samples, either biopsy or TURP, were obtained and centrally reviewed to obtain consistent pathological evaluation to the current standards. Drawing from this cohort, we carried out genome-wide CNA analysis by array-based comparative genomic hybridization (aCGH) of 107 biopsies or TURP samples from the TAPG1 cohort, as tissue availability is limited for much of the full cohort. The subset of cases used for CNA analysis, which make up our conservative treatment CNA cohort, have similar clinical characteristics to the full TAPG1 cohort, including median diagnosis age, baseline PSA, hormonal treatment, and clinical stage, with the exception of higher Gleason score distribution, likely due to selection for cases with sufficient DNA for analysis (*Supplementary file 2*). As expected for a cohort not subject to PSA screening, the patients are older and have higher grade at diagnosis than is typical for contemporary US cohorts. Among the cohort, 47 patients developed metastasis and 43 died of prostate cancer. The median follow-up time for survivors was 10.3 years from diagnosis.

The copy number alteration landscape of the conservative treatment cohort revealed canonical copy number alterations of prostate cancer, including gain of chromosome 8q and losses on chromosomes 6 p, 8 p, 13q and 16 p, though with lower frequency than seen in prostate cancer cohorts analyzed by our group (MSKCC cohort) (*Taylor et al., 2010*) and TCGA (*Cancer Genome Atlas Research Network, 2015*) (*Figure 1a*). The percentage of the cancer genome showing copy number changes, termed tumor CNA burden (TCB), is similar between the conservative treatment CNA cohort and other cohorts (*Figure 1b*), with a mean tumor CNA burden of 5.7% (median 1.5%, IQR 0.05–8.5%) compared to 5.2% (median 3.0%, IQR 0.04–6.9%) for the 2010 MSKCC primary prostate cancer cohort (*Taylor et al., 2010*) and 4.0% (median 0.7%, IQR 0.08–5.1%) for the 2014 MSKCC

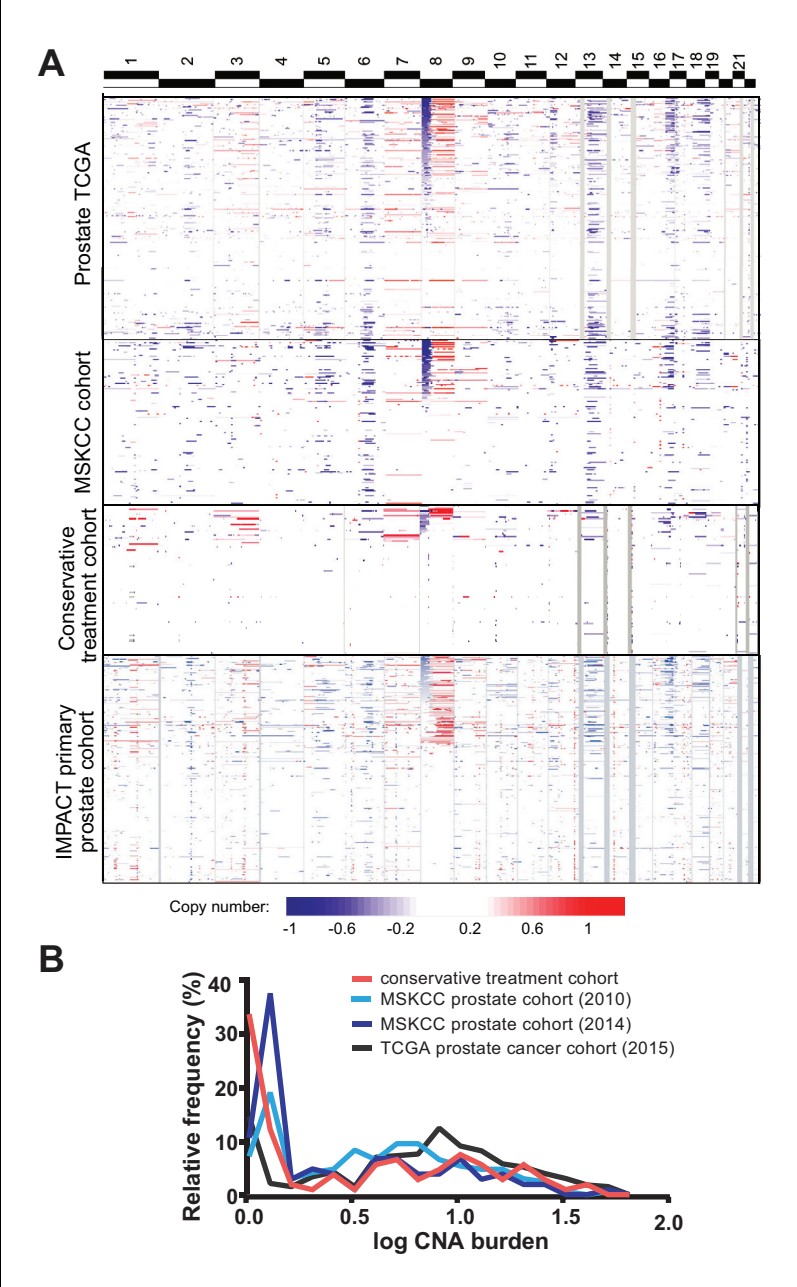

**Figure 1.** Tumor copy number landscape of conservatively treated primary prostate cancer, compared to other primary prostate cancer cohorts. (a) Heat map of copy number alterations in conservative treatment CNA cohort, as well as TCGA, MSKCC, and IMPACT primary prostate cancer cohorts. (b) Frequency distribution of CNA burden, as log of percentage of genome copy number altered, for the conservative treatment prostate cancer cohort and three other primary prostate cancer cohorts.
DOI: https://doi.org/10.7554/eLife.37294.003

primary prostate cancer cohort (*Hieronymus et al., 2014*). The tumor CNA burden of the conservative treatment CNA cohort is, however, somewhat lower than the 8.7% average tumor CNA burden of the TCGA prostate cohort (*Cancer Genome Atlas Research Network, 2015*) (mean 8.7%, median 6.2%, IQR 1.7–11.9%).

## Tumor CNA burden is prognostic for prostate cancer-specific death

Since tumor CNA burden is associated with prostate cancer recurrence and metastasis in prostatectomy cohorts (*Taylor et al., 2010*; *Hieronymus et al., 2014*), we sought to determine whether tumor CNA burden was prognostic for cancer-specific death in biopsies of conservatively treated prostate cancer. In our conservative treatment CNA cohort, we find that tumor CNA burden as a continuous variable is significantly associated with prostate cancer-specific death (per 5% tumor CNA burden, HR 1.49; 95% CI 1.30, 1.70; p<0.0001; *Table 1*). Greater tumor CNA burden correlates with an increase in death from disease compared to a lower tumor CNA burden (*Figure 2a*). The risk of death due to prostate cancer within 5 years of diagnosis increases with tumor CNA burden over the majority of the tumor CNA burden distribution (*Figure 2b*). For example, the 5 year risk of death due to prostate cancer would be 13% for patients with a 2% tumor CNA burden and 28% for patients with a 10% tumor CNA burden (*Figure 2b*). Tumor CNA burden may therefore serve as a prognostic factor for cancer-specific death in patients who undergo increasingly common conservative treatment approaches.

We next asked whether tumor CNA burden was associated with outcome after adjusting for established prognostic variables, including Gleason sum score and the UCSF Cancer of the Prostate Risk Assessment (CAPRA) score (*Cooperberg et al., 2005*; *Brajtbord et al., 2017*) which combines PSA, Gleason score, percentage positive biopsy cores, clinical stage, and age (*Figure 2c*). Tumor CNA burden is significantly associated with cancer-specific death even after adjusting for biopsy Gleason score (per 5% tumor CNA burden, HR 1.44; 95% CI 1.24, 1.67; p<0.0001) or CAPRA score (per 5% tumor CNA burden, HR 1.44; 95% CI 1.24, 1.68; p<0.0001) (*Table 1*, *Figure 2c*). The addition of tumor CNA burden into the model with the CAPRA score increased Harrell's concordance index from 0.756 to 0.805 for cancer-specific survival in our cohort of men with conservatively treated prostate cancer.

## Tumor CNA burden is prognostic for cancer-free and overall survival in multiple cancer types

Large, clinically annotated cancer genomic efforts such as TCGA now provide an opportunity to examine whether CNA burden is prognostic for primary cancer outcomes across many cancer types. In the TCGA primary prostate cancer cohort (*Cancer Genome Atlas Research Network, 2015*), tumor CNA burden is significantly associated with biochemical recurrence individually (p<0.0001; per 5% tumor CNA burden, HR = 1.27; 95% CI, 1.13, 1.42) and after adjustment for Gleason score and mutation burden (p=0.015; per 5% tumor CNA burden, HR = 1.18; 95% CI, 1.03, 1.35), validating our findings from other prostate cancer cohorts (*Figure 2c*, *Figure 2—figure supplement 1*, *Table 2*). There were insufficient deaths in this cohort to analyze survival. CNA burden was still significantly associated with biochemical recurrence after adjusting for tumor sample purity determined by ABSOLUTE (p<0.003; per 5% CNA burden, HR = 1.22; 95% CI, 1.07, 1.40; *Table 2*). Since tumor CNA burden could potentially reflect simply the prognostic significance of aneuploidy as determined by cytometric DNA index in various cancers (*Walther et al., 2008*; *Danielsen et al., 2016*), we examined the tumor CNA burden in a multivariable model together with ploidy. Ploidy, generated by CLONET and previously published for this cohort, estimates the average DNA index of the tumor cells (*Carter et al., 2012*; *Prandi et al., 2014*). Tumor CNA burden was associated with recurrence

**Table 1.** Tumor CNA burden is associated with prostate cancer-specific death in conservative treatment cohort independent of Gleason sum score and CAPRA score.

Cox Regression model assessing the association between CNA burden (per 5%) and cancer specific survival. N = 107*

| Model | HR | 95% CI | P-value |
|---|---|---|---|
| Univariate, tumor CNA burden | 1.49 | 1.30, 1.70 | <0.0001 |
| Multivariable – adjusting for Gleason sum (≤6, 7, ≥8) | 1.44 | 1.24, 1.67 | <0.0001 |
| Multivariable – adjusting for UCSF-CAPRA score utilizing multiple imputation | 1.44 | 1.24, 1.68 | <0.0001 |
| Multivariable – adjusting for UCSF-CAPRA score without utilizing multiple imputation | 1.57 | 1.29, 1.92 | <0.0001 |

* N = 60 (excludes 47 patients with unknown stage)

DOI: https://doi.org/10.7554/eLife.37294.010

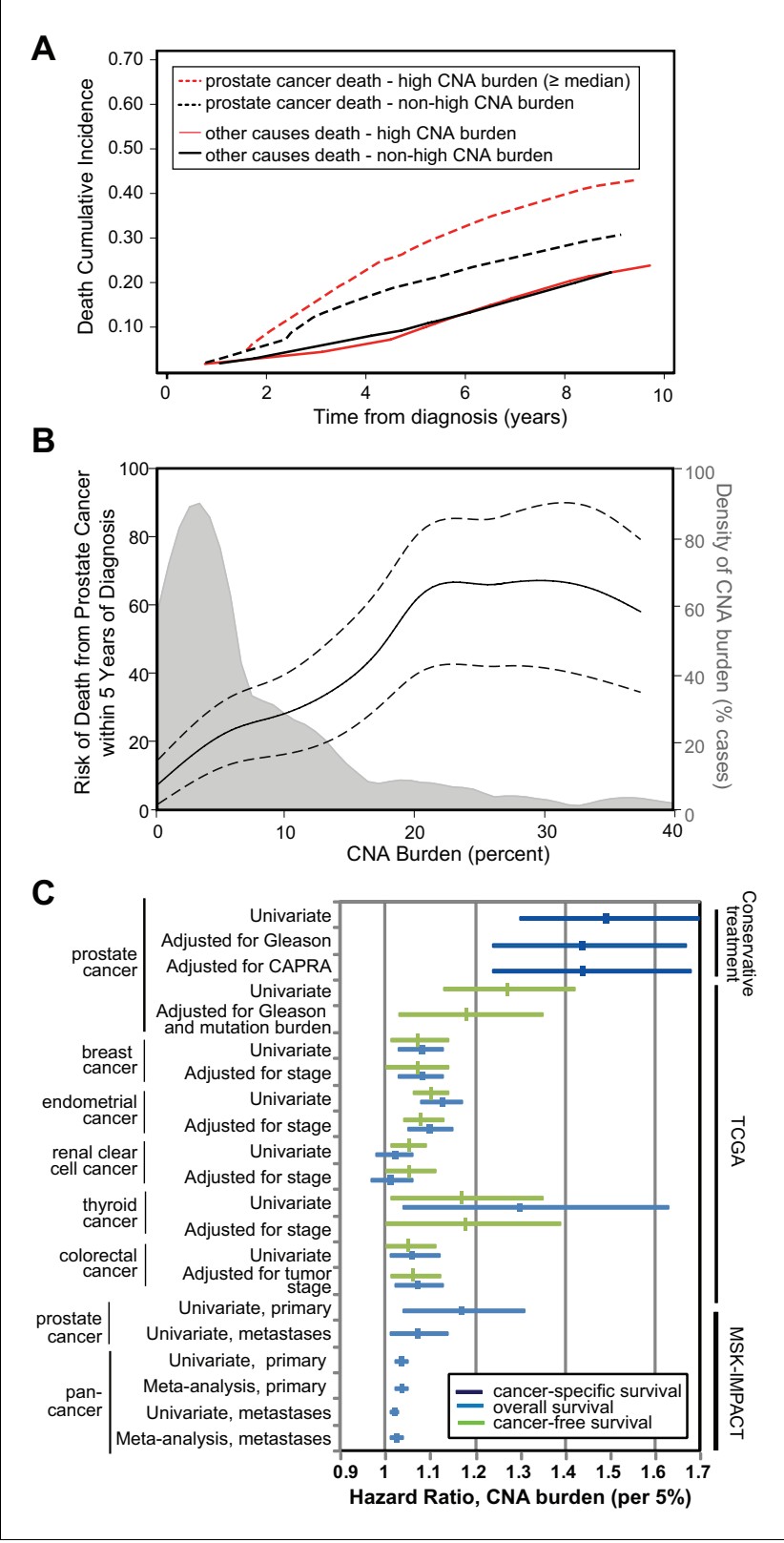

**Figure 2.** Tumor copy number alteration burden is associated with death from prostate cancer in conservatively treated patients. (a) Cumulative Incidence of death from disease (dashed lines) and death from other causes (solid lines) based in cases with high CNA burden (red lines, CNA Burden greater than or equal to the median CNA burden of this cohort, 1.48) or non-high CNA burden (black lines, CNA Burden < median). (b) Risk for death from

*Figure 2 continued*

prostate cancer within 5 years of diagnosis. Univariate risk for 5 year prostate cancer-specific death, calculated by locally weighted Kaplan–Meier estimates (solid black line) with 95% confidence interval (dashed black lines) overlaid on the distribution of CNA burden (gray). (c) Association of tumor CNA burden with available cancer outcomes in the conservative treatment primary prostate cancer TAPG1 cohort, TCGA primary cancer cohorts, and the MSK-IMPACT clinical sequencing prostate and pan-cancer cohorts of primary and metastatic tumors. Forest plot of hazard ratio (per 5% CNA burden) with 95% confidence interval shown for cancer-specific mortality (dark blue), overall mortality (light blue), and cancer recurrence (green). Supplementary Tables and Figures.

DOI: https://doi.org/10.7554/eLife.37294.004

The following figure supplements are available for figure 2:

**Figure supplement 1.** Kaplan-Meier plot of biochemical recurrence in TCGA primary prostate cohort.
DOI: https://doi.org/10.7554/eLife.37294.005

**Figure supplement 2.** Tumor CNA burden in multiple cancers is associated with disease free survival and overall survival.
DOI: https://doi.org/10.7554/eLife.37294.006

**Figure supplement 3.** Correlation between CNA burden from IMPACT targeted sequencing assay and whole exome sequencing (WES) of same samples, pan-cancer.
DOI: https://doi.org/10.7554/eLife.37294.007

**Figure supplement 4.** Tumor CNA burden in primary prostate cancer is prognostic for overall survival when assayed by clinically approved sequencing panel.
DOI: https://doi.org/10.7554/eLife.37294.008

**Figure supplement 5.** Forest Plot of Hazard Ratios (individual and pooled) for meta-analysis assessing the association between tumor CNA burden and overall survival in (a) primary cancer and (b) patients with metastatic cancer in the pan-cancer IMPACT cohort.
DOI: https://doi.org/10.7554/eLife.37294.009

independent of tumor ploidy (p=0.002; per 5% tumor CNA burden, HR = 1.32; 95% CI 1.11, 1.56; *Table 2*). Moreover, for a multivariable model that includes tumor CNA burden, Gleason grade, and mutation burden, the Harrell's C-index is 0.691. In contrast, the C-index for a model including ploidy instead of tumor CNA burden is only 0.606, indicating that a model with clinical factors and ploidy does not perform as well as a model with the same clinical factors and tumor CNA burden.

The prognostic significance of tumor CNA burden in prostate cancer led us to ask whether tumor CNA burden is prognostic in other cancer types. Towards this end, we examined published TCGA cohorts for multiple cancer types with available disease-free survival and overall survival data, including breast (*Ciriello et al., 2015*), endometrial (*Cancer Genome Atlas Research Network et al., 2013*), renal clear cell (*Cancer Genome Atlas Research Network, 2013*), thyroid (*Cancer Genome Atlas Research Network, 2014*), and colorectal (*Cancer Genome Atlas Network, 2012*) cancers. We found that tumor CNA burden is associated with recurrence (disease-free survival) in these cancer types (*Figure 2c*, *Figure 2—figure supplement 2*, *Table 2*). This association between tumor CNA burden and lower disease-free survival was independent of disease stage in all cancer types except colorectal cancer, where the association was independent of tumor stage but not disease stage (*Table 2*). In addition to lower disease-free survival, higher tumor CNA burden was also significantly associated with lower overall survival in breast, endometrial, thyroid, and colorectal cancer (*Table 2*). This association with overall survival was independent of disease stage in breast and endometrial cancer and independent of tumor stage in colorectal cancer (*Table 2*). There were insufficient cases of thyroid cancer with stage data for this analysis. In summary, tumor CNA burden is prognostic for recurrence and/or overall survival in multiple cancer types beyond prostate cancer, including breast, endometrial, colorectal, renal clear cell, and thyroid cancer.

## Tumor CNA burden determined by clinical targeted sequencing of primary and metastatic tumors is prognostic for survival

We next wanted to determine whether CNA burden's prognostic associations could be observed using panel-based targeted sequencing assays that are increasingly entering clinical use, in contrast to CGH array-based determination of tumor CNA burden. The Memorial Sloan Kettering-Integrated Mutation Profiling of Actionable Cancer Targets (MSK-IMPACT) assay is a clinical laboratory improvement amendments (CLIA)-certified sequencing-based assay (*Cheng et al., 2015*) of several

**Table 2.** Tumor CNA burden is associated with recurrence and overall survival independent of disease stage in multiple cancer types

| Cohort | Model | Disease free Time | | | | Overall Survival | | | |
|---|---|---|---|---|---|---|---|---|---|
| | | Cases | HR | 95% CI | P | Cases | HR | 95% CI | P |
| Prostate cancer TCGA | Tumor CNA burden, per 5% tumor CNA burden, univariate | 280 | 1.27 | 1.13, 1.42 | <0.0001 | Insufficient events | | | |
| | Tumor CNA burden, per 5% tumor CNA burden, adjusted for Gleason grade and mutation burden | 279 | 1.18 | 1.03, 1.35 | 0.015 | | | | |
| | Tumor CNA burden, per 5% tumor CNA burden, adjusted for purity (ABSOLUTE)* | 243 | 1.22 | 1.07, 1.40 | 0.003 | | | | |
| | Tumor CNA burden, per 5% tumor CNA burden, adjusted for ploidy | 243 | 1.32 | 1.11, 1.56 | 0.002 | | | | |
| Breast cancer TCGA | Tumor CNA burden, per 5% tumor CNA burden, univariate | 709 | 1.07 | 1.01, 1.14 | 0.028 | 794 | 1.08 | 1.03, 1.13 | 0.0005 |
| | Tumor CNA burden, per 5% tumor CNA burden, Multivariable, adjusted for disease stage | 695 | 1.07 | 1.00, 1.14 | 0.049 | 777 | 1.08 | 1.03, 1.13 | 0.002 |
| Endometrial Cancer TCGA | Tumor CNA burden, per 5% tumor CNA burden, univariate | 496 | 1.10 | 1.06, 1.14 | <0.0001 | 536 | 1.13 | 1.08, 1.17 | <0.0001 |
| | Tumor CNA burden, per 5% tumor CNA burden, multivariable, adjusted for disease stage | 496 | 1.08 | 1.04, 1.13 | <0.0001 | 536 | 1.10 | 1.05, 1.15 | <0.0001 |
| Renal clear cell cancer TCGA | Tumor CNA burden, per 5% tumor CNA burden, univariate | 425 | 1.05 | 1.01, 1.09 | 0.028 | 525 | 1.02 | 0.98, 1.06 | NS (0.4) |
| | Tumor CNA burden, per 5% tumor CNA burden, multivariable, adjusted for disease stage | 423 | 1.05 | 1.00, 1.11 | 0.035 | 522 | 1.01 | 0.97, 1.06 | NS (0.7) |
| Thyroid cancer TCGA | Tumor CNA burden, per 5% tumor CNA burden, univariate | 483 | 1.17 | 1.01, 1.35 | 0.033 | 497 | 1.30 | 1.04, 1.63 | 0.021 |
| | Tumor CNA burden, per 5% tumor CNA burden, multivariable, adjusted for disease stage | 481 | 1.18 | 1.00, 1.39 | 0.048 | Insufficient events | | | |
| Colorectal cancer TCGA | Tumor CNA burden, per 5% tumor CNA burden, univariate | 512 | 1.05 | 1.00, 1.11 | 0.037 | 587 | 1.06 | 1.01, 1.12 | 0.012 |
| | Tumor CNA burden, per 5% tumor CNA burden, multivariable, adjusted for disease stage | 496 | 1.03 | 0.98, 1.09 | NS (0.3) | 567 | 1.03 | 0.97, 1.09 | NS (0.3) |
| | Tumor CNA burden, per 5% tumor CNA burden, multivariable, adjusted for tumor stage | 511 | 1.06 | 1.01, 1.12 | 0.028 | 585 | 1.07 | 1.02, 1.13 | 0.009 |

*Result differed with similar adjustment in IMPACT prostate cancer cohort utilizing FACETS, see **Supplementary file 3**.

DOI: https://doi.org/10.7554/eLife.37294.011

hundred cancer genes and 1042 common single nucleotide polymorphisms (SNPs) that has been used to profile 504 prostate tumors (*Abida et al., 2017*) and more than ten thousand tumors across other cancer types (*Zehir et al., 2017*). The IMPACT assay identifies both somatic point mutations and copy number alterations in the genes included in the panel. Overall copy number burden is calculated across the whole genome (*Figure 1a*) using segmentation derived from a combination of the profiled SNPs to provide low resolution copy number data and the genes sequenced in the panel (*Cheng et al., 2015*; *Abida et al., 2017*; *Zehir et al., 2017*). To address the possibility that CNA burden from the IMPACT panel might differ from that derived from more comprehensive sequencing, we directly compared CNA burden calculations from 1005 tumors that were profiled using both IMPACT and whole exome sequencing. CNA burden determined by the two methods were highly correlated (p-value<0.0001, rho = 0.88, n = 1005), indicating that CNA burden is not significantly affected by the reduced resolution in moving from whole exome to targeted panel sequencing (*Figure 2—figure supplement 3*).

We find that tumor CNA burden assayed by targeted clinical sequencing is significantly associated with overall survival in primary prostate tumors (per 5% tumor CNA burden, HR = 1.17; 95% CI, 1.04, 1.3; p=0.007; *Table 3*, *Figure 2c*, *Figure 2—figure supplement 4*) in the IMPACT prostate cohort (*Abida et al., 2017*). As clinical sequencing assays such as MSK-IMPACT are principally used in the metastatic patient population, the IMPACT cohorts also provide an opportunity to investigate the prognostic significance of tumor CNA burden in late stage disease. We find that tumor CNA

**Table 3.** Tumor CNA burden determined by clinically approved sequencing panel is associated with overall survival in primary and metastatic tumors

| | Overall Survival | | | | | |
| | Primary tumors | | | Metastatic tumors | | |
| Model | HR | 95% | P | HR | 95% | P |
|---|---|---|---|---|---|---|
| Prostate Cancer[*,†] | | | | | | |
| Univariate, tumor CNA burden, per 5% | 1.17 | 1.04, 1.31 | 0.007 | 1.07 | 1.01, 1.14 | 0.020 |
| Multivariable Tumor CNA burden, per 5% Mutation burden (per mutation) | 1.11 1.22 | 0.98, 1.26 1.12, 1.33 | 0.10 <0.0001 | 1.08 1.05 | 1.02, 1.15 1.02, 1.08 | 0.011 0.001 |
| Multivariable Tumor CNA burden, per 5% TP53 CN loss or mutation | 1.17 4.12 | 1.04, 1.31 2.02, 8.41 | 0.007 <0.0001 | 1.06 1.24 | 1.00, 1.13 0.76, 2.02 | NS (0.069) NS (0.4) |
| Multivariable Tumor CNA burden, per 5% RB1 CN loss or mutation | 1.15 3.24 | 1.02, 1.30 0.70, 14.98 | 0.026 NS (0.13) | 1.06 1.68 | 0.99, 1.13 0.94, 2.99 | NS (0.091) NS (0.080) |
| Multivariable Tumor CNA burden, per 5% PTEN CN loss or mutation | 1.17 2.38 | 1.04, 1.32 1.03, 5.51 | 0.008 0.042 | 1.07 1.15 | 1.01, 1.14 0.70, 1.89 | 0.023 NS (0.6) |
| Pan- Cancer | | | | | | |
| Univariate, tumor CNA burden, per 5%[‡§] | 1.04 | 1.02, 1.05 | <0.0001 | 1.02 | 1.01, 1.03 | 0.005 |
| Univariate, mutation burden (per five units)[‡§] | 0.98 | 0.97, 1.00 | NS (0.072) | 0.99 | 0.97, 1.01 | NS (0.4) |
| Meta-analysis, tumor CNA burden (per 5%)[#] | 1.04 | 1.02, 1.05 | <0.0001** | 1.02 | 1.01, 1.04 | 0.005[††] |
| Meta-analysis, tumor CNA burden (per 5%), excluding outlier cancer types[‡‡] | 1.05 | 1.03, 1.07 | <0.0001[§§] | 1.03 | 1.01, 1.04 | 0.002[##] |

*Prostate primary tumors: patient n = 261 for all models except multivariable model with mutation burden, where n = 227; event n = 33; median follow-up time for survivors 40 (IQR 25,81) months.

†Prostate metastatic tumors: patient n = 216 for all models except multivariable model with mutation burden, where n = 205; event n = 80; median follow-up time for survivors 59.5 (IQR 32, 129) months.

‡Pan-cancer primary tumors, univariate models: patient n = 6610, event n = 1535, median follow-up time for survivors 24 (IQR 11, 61) months

§Pan-cancer metastatic tumors, univariate models: patient n = 4864, event n = 1467, median follow-up time for survivors 51 (IQR 23, 109) months.

#Pan-cancer meta-analysis, among ten most prevalent cancer types: primary tumor patient n = 4863, metastatic tumor patient n = 3676. Estimates are based on overall fixed effects.

**p-value corresponds with test of effects size. Chi-square test for heterogeneity p-value=0.003.

††p-value corresponds with test of effects size. Chi-square test for heterogeneity p-value=0.024.

‡‡Exclusion of cancer types to reduce heterogeneity: primary tumor patient n = 3887, metastatic tumor patient n = 3098. Estimates are based on overall fixed effects.

§§Excluding pancreatic and colorectal cancer, test of effects size p-value. Chi-square test for heterogeneity p-value=0.3.

##Excluding pancreatic and prostate cancer, test of effects size p-value. Chi-square test for heterogeneity p-value=0.8.

DOI: https://doi.org/10.7554/eLife.37294.012

burden of metastatic prostate tumors assayed by clinical sequencing is also significantly associated with survival (per 5% tumor CNA burden, HR = 1.07; 95% CI, 1.01, 1.14; p=0.020; *Table 3*, *Figure 2c*, *Figure 2—figure supplement 4*).

Since clinical sequencing assays also provide point mutation information for several hundred cancer genes, we asked if tumor CNA burden is prognostic after adjusting for known prostate cancer driver alterations. In separate multivariable regression models adjusting for *TP53*, *RB1*, or *PTEN* loss and/or mutation, tumor CNA burden is still associated with overall survival independent of these alterations in primary prostate tumors (*Table 3*). In metastatic tumors, these specific gene mutations do not reach prognostic significance when combined with tumor CNA burden (*Table 3*). Notably, tumor CNA burden remains significant in metastatic tumors after adjusting for overall tumor mutation burden (per 5% tumor CNA burden, HR = 1.08; 95% CI = 1.02, 1.15; p=0.011; *Table 3*).

As targeted clinical sequencing is applied to a wide range of cancer types, we expanded our survival analysis to a pan-cancer cohort, consisting of 6610 primary tumors and 4864 metastatic tumors

across 53 cancer types assayed by MSK-IMPACT sequencing panel (Materials and methods and *Supplementary file 2*). We find that tumor CNA burden is prognostic for overall survival pan-cancer in primary tumors (p<0.0001; per 5% tumor CNA burden, HR = 1.04; 95% CI, 1.02, 1.05) and in metastatic tumors (p=0.005; per 5% tumor CNA burden, HR = 1.02; 95% CI, 1.01, 1.03) in a univariate analysis of these pan-cancer cohorts (*Table 3*, *Figure 2c*). Tumor CNA burden is also prognostic for cancer-specific death in the metastatic tumor cohort (p=0.026; per 5% tumor CNA burden, HR = 1.05; 95% CI, 1.01, 1.10). Adjustment for sample tumor purity determined by FACETS (*Shen and Seshan, 2016*) found that CNA burden was still significantly associated with overall survival in primary tumors in the pan-cancer analysis and approached significance for metastatic tumors (p=0.06; *Supplementary file 3*), though purity-adjusted CNA burden was no longer significantly associated with overall survival in the prostate tumor subsets (*Supplementary file 3*). Adjustment for sample tumor purity determined by FACETS (*Shen and Seshan, 2016*) found that CNA burden was still significantly associated with overall survival in primary tumors in the pan-cancer analysis approached significance for metastatic tumors (p=0.06; *Supplementary file 3*), though purity-adjusted CNA burden was no longer significantly associated with overall survival in the prostate tumor subsets (*Supplementary file 3*). Tumor mutation burden (TMB), in contrast to tumor CNA burden, was not associated with overall survival or cancer-specific survival (p=0.4 and p>0.9, respectively; *Table 3*).

Since the pan-cancer prognostic significance of tumor CNA burden is likely to be influenced by the distribution of cancer types within the IMPACT cohorts, we stratified the primary and metastatic pan-cancer IMPACT cohorts by their ten most prevalent cancer types, which make up nearly three-quarters of the cohort (*Supplementary file 2*). A multivariable Cox model was used for each cancer type to adjust for mutation burden and extract the effect size, which was then entered into a meta-analysis. After stratifying by cancer type, the CNA burden of primary tumors measured by the MSK-IMPACT assay is still significantly associated with death (overall fixed effects HR = 1.04; 95% CI 1.02, 1.05; test of effects size p<0.0001; *Table 3*; *Figure 2c*). Similarly, metastatic tumor CNA burden was associated with death (overall fixed effects HR = 1.02; 95% CI 1.01, 1.04; test of effects size p=0.005; *Table 3*; *Figure 2c*).

A closer look at the pan-cancer analysis reveals statistically significant heterogeneity in the relationship between tumor CNA burden and survival across tumor types (p=0.003 and p=0.024 in primary and metastatic tumor cohorts respectively, *Figure 2—figure supplement 4*). In primary tumors, heterogeneity appears to be driven by colorectal and pancreatic cancers, where an inverse association between tumor CNA burden and death is seen (*Figure 2—figure supplement 5a*). After excluding colorectal and pancreatic cancers, heterogeneity is no longer statistically significant (overall fixed effects HR = 1.05; 95% CI 1.03, 1.07; test of effects size p<0.0001; test for heterogeneity p=0.3; *Figure 2—figure supplement 5a*). In metastatic tumors, two outlying cancer types drive this heterogeneity: pancreatic cancer, which shows the same inverse association of tumor CNA burden with death as in primary pancreatic tumors, and prostate, which shows the opposite effect (*Figure 2—figure supplement 5b*). Exclusion of either cancer type eliminates the significant heterogeneity in effects size, such that higher tumor CNA burden is associated with increased death in the remaining homogenous set of cancer types (overall fixed effects HR = 1.03; 95% CI 1.01, 1.04; test of effects size p=0.002; test for heterogeneity p=0.8, *Figure 2—figure supplement 5b*). These results indicate that tumor CNA burden can have differing levels of prognostic effect depending on the cancer type, while a core set of cancer types show a statistically similar association between overall survival and tumor CNA burden assayed by targeted sequencing. More generally, we find that tumor CNA burden determined by a clinically-certified sequencing panel is associated with overall and disease-specific mortality in a large multi-cancer population, including in patients with metastatic cancer where clinical sequencing is increasingly applied.

## Discussion

Many specific genes altered by CNA have been associated with cancer outcomes (*Liang et al., 2016*; *Wang et al., 2016*; *Nibourel et al., 2017*), however the relationship between outcome and the overall level of CNA harbored by a tumor is less well studied. Here we expanded on our previous work showing that tumor CNA burden is associated with recurrence in surgically treated primary prostate cancer (*Taylor et al., 2010*; *Hieronymus et al., 2014*) by showing a significant association

with death from prostate cancer, including in conservatively treated patients where the tumor CNA burden measurement was made from biopsies. Importantly, this association remains significant even after adjusting for Gleason score or for CAPRA score, demonstrating that CNA burden is independent of previously identified associations with these measures of cancer pathology or disease state. Thus, tumor CNA burden assessment from prostate biopsies could have a role in deciding between surgery and surveillance for men at the low end of intermediate risk. Conversely, it may also have role in men at high risk where multimodal treatment may be needed.

An unanticipated outcome of our analysis beyond prostate cancer is the prognostic role of tumor CNA burden across a range of tumor types. The pan-cancer tumor CNA burden association is significant but also heterogeneous depending on cancer type. Recent work has similarly found that the presence of any CNA, regardless of gene identity, is associated with overall and event-free survival in pediatric AML (*Vujkovic et al., 2017*) and that the percentage of subclonal CNAs but not subclonal somatic point mutations is associated with overall survival in non-small cell lung cancer (*Jamal-Hanjani et al., 2017*). Moreover, survival time was associated with a CNA signature derived from supervised analyses in prostate cancer and extended to breast and lung cancer (*Pearlman et al., 2018*). Prognostic individual CNAs or sets of CNAs, as opposed to the broader measure of genome-wide CNA level examined here may be specific to individual cancer types, whereas we have demonstrated the prognostic potential of a generalized measure of overall copy number dysregulation. Further work will be needed to address the trade-offs between generalizability of CNA burden and discriminatory power. In addition, it will be important to investigate whether the prognostic associations of CNA burden from the pan-cancer analysis are independent of known cancer- or subtype-specific prognostic factors, such as ER receptor status in breast cancer, ultra- and hypermutated (POLE and MSI+) status in endometrial cancer and MSI-positive/CIN-negative status in colorectal cancer (*Walther et al., 2008*).

We find it notable that tumor CNA burden assessment using a targeted sequencing can serve as a surrogate for tumor CNA burden calculated using more comprehensive genomic assays such as array CGH. With the proliferation of different clinical sequencing panels for mutation detection, it will be of interest to see how much resolution, depth, and coverage can be reduced and still retain the prognostic association of CNA burden; future work in this area will also need to incorporate the predictive clinical utility of the point mutation data to address the multimodal uses of clinical sequencing assays. Another important variable is tumor purity. The prognostic significance of CNA burden can be affected by sample tumor purity, with purity being independently associated with outcome. The effect of purity on the association between CNA burden and outcome appears complex and may be influenced by the analysis platform, cancer type, and outcome type. For example, pan-cancer CNA burden from clinical sequencing panel remained prognostic for survival after purity adjustment in primary tumors and was just below significance for metastatic tumors, though the CNA burden of the prostate tumor subset assayed by IMPACT sequencing panel did not. However, the CNA burden of prostate tumors assayed by SNP array showed continued association with recurrence after adjustment for purity. Tumor purity alone was also independently associated with survival, revealing a complex interaction between these tumor features that will need further exploration. As targeted sequencing moves from tumor samples to liquid biopsy in the form of cell-free DNA (cfDNA) (*Heitzer et al., 2016*; *Xia et al., 2015*; *Hyman et al., 2017*), it will be important to determine whether tumor CNA burden determined by analysis of cfDNA has similar prognostic utility as that determined by direct analysis of tumor DNA. There is already some evidence this may be possible, as the summed CNA level of the most highly copy number altered genes assayed from whole genome sequencing of cfDNA in twenty metastatic prostate cancer patients correlated with overall survival (*Xia et al., 2015*). As sequencing costs continue to drop and computational power improves, it would be interesting to investigate low pass whole genome sequencing as an alternative approach for determining tumor CNA burden that provides complete genome coverage.

Another interesting feature of the association of tumor CNA burden with outcome demonstrated here is that it has prognostic significance independent of tumor mutation burden (TMB). This is consistent with recent work in glioblastoma, breast, lung, and ovarian cancer showing that CNA-derived signatures have more prognostic power than somatic point mutation-based signatures, as measured by concordance index (*Gómez-Rueda et al., 2015*). Thus, tumor CNA burden could complement clinical analyses of actionable driver mutations using a single panel-based sequencing assay.

The prognostic significance of tumor CNA burden raises intriguing questions regarding the underlying biology. Tumor CNA burden may be a simple measure that correlates with the extent of oncogenic driver alterations. Yet, we show that tumor CNA burden retains its prognostic significance after adjustment for a number of known oncogenic alterations in primary prostate cancer, including PTEN loss associated with increased tumor CNA burden (*Castro et al., 2015*; *Williams et al., 2014*). In metastatic tumors, combining tumor CNA burden with TP53 or RB1 loss in multivariable analyses renders both slightly below conventional significance thresholds, raising the possibility of biological interplay between these genes (particularly TP53) and subsequent copy number alteration that develops during tumor evolution. Further, the prognostic associations of tumor CNA burden are independent of tumor ploidy, which suggests that tumor CNA burden may not simply reflect aneuploidy, defined as abnormal DNA content (*Danielsen et al., 2016*). It is also possible that tumor CNA burden captures prognostic information about currently unidentified driver alterations and/or the rate of ongoing CNA within a tumor that may generate additional driver alterations, including those reflecting intratumoral heterogeneity, thereby affecting outcome. Ongoing work by others has begun to develop genomic methods for identifying mechanisms of somatic CNA (*Wala et al., 2017*); and identify prognostic CNA signatures and the mechanisms underlying the component CNA (*Macintyre et al., 2018*). Ultimately, the biology underlying the significant association of tumor CNA burden with multiple cancer outcomes will be a fruitful area for future investigation.

## Materials and methods

### aCGH copy number analysis of conservative-treatment TAPG cohort.

Of the TAPG1 cohort (*Cuzick et al., 2006*), FFPE prostate tumor tissue from 180 patients was macrodissected from slides. DNA was isolated (Agilent FFPE DNA isolation for aCGH protocol) and quantified by picogreen-based quantification. 107 cases yielded greater than 500 ug DNA and were analyzed by Agilent 180K human CGH arrays (Agilent, 4 × 180K G4449A arrays, per manufacturer's instructions). Copy number data from patients in the TAPG copy number cohort were quantified, normalized, segmented, and analyzed with RAE, as previously described (*Taylor et al., 2010*; *Hieronymus et al., 2014*). The conservative treatment TAPG copy number cohort array data was deposited in NCBI GEO under accession number GSE103665 (Gene Expression Omnibus, RRID: SCR_007303).

Tumor CNA burden (tumor CNA burden) was analyzed as percent CNA burden, defined as the length of the genome altered by copy number alteration multiplied by 100. For regression analyses, tumor CNA burden was scaled as per five percent so that the estimates of our hazard ratios were more interpretable. All statistical analyses were performed using Stata 13 (RRID:SCR_012763, StataCorp, College Station, TX).

### TAPG copy number cohort statistical analyses

For Cox regression analyses, the primary aim was to determine whether tumor CNA burden is associated with cancer specific survival (CSS). First, we assessed whether there was an association between tumor CNA burden and CSS by utilizing a univariate Cox model, censoring patients who did not die at the date of their last follow-up and patients who died of other causes at their death date. Secondly, in order to assess whether there is information from tumor CNA burden over and above biopsy Gleason score, we utilized a multivariable Cox model, adjusting for biopsy Gleason sum categorized as $\leq$6, 7, and $\geq$8. Finally, to assess whether there is an association between tumor CNA burden and CSS after accounting for the preoperative predictors of CSS, we utilized a multivariable Cox model, adjusting for the UCSF-CAPRA score, a preoperative risk score calculated by incorporating the patient's age at diagnosis, PSA at diagnosis, primary and secondary Gleason score at biopsy and clinical tumor stage. As percent of positive biopsy cores was not available for the cohort, a modified CAPRA score was utilized not incorporating this information. Among our cohort of 107 patients, 47 patients were missing clinical tumor stage; multiple imputation was used to impute the missing values. Statistical analyses were performed utilizing the measured and imputed values combined across 10 imputations using Rubin's method. Furthermore, to evaluate the discriminative accuracy of the model including tumor CNA burden, we calculated bootstrap optimism-corrected Harrell's C-index. It should be noted that the discrimination of the CAPRA score is lower in the

TAPG1 conservative treatment CNA cohort than seen in some other prostate cancer cohorts, and this may impact the degree to which tumor CNA burden increases the concordance index. All data used for these analyses are available in *Supplementary file 4*.

For illustrative purposes, we utilized competing risk methods to estimate the probability of death from prostate cancer in the setting of death from other causes. Cumulative incidence was shown for patients who died from prostate cancer, or died from other causes, stratified on tumor CNA burden in relation to the median tumor CNA burden among the cohort, using the *stcompet* command in Stata.

## Statistical analyses of IMPACT cohorts

For analysis of the prostate cancer MSK-IMPACT cohort (*Abida et al., 2017*), the published cases were analyzed by Cox regression for association between overall survival and tumor CNA burden (*Supplementary file 5* and *6*). The IMPACT cases were separated into groups consisting of primary tumors or metastatic tumors, including loco-regional, non-resistant to treatment, and treatment resistant, though primary tumor samples include cases sampled after metastatic spread. Among our primary and metastatic IMPACT prostate cancer cohorts, we excluded men with unknown overall survival status and unknown time until overall survival status, leaving us with a final cohort of 261 and 216 men, respectively. Among these two groups of patients, we assessed the association between tumor CNA burden and overall survival using a univariate Cox model. Multivariable Cox models were then used to determine whether the association between tumor CNA burden and overall survival remained after accounting for purity determined by FACETS (*Shen and Seshan, 2016*), the overall point mutation burden, or specific somatic gene alterations (shallow or deep copy number loss or mutation) occurring in prostate cancer (*BRCA1, BRCA2, ATM, TP53, RB1*, and *PTEN*), using separate models for each alteration. As the overall point mutation burden was not available for all patients, 34 patients with primary prostate cancer and 11 patients with metastatic prostate cancer were excluded from this portion of the analysis in their respective cohorts.

For analysis of our pan-cancer IMPACT cohort (MSK-IMPACT cohort (*Zehir et al., 2017*) and additionally accrued IMPACT samples), outcome data at time of analysis, mutation burden, and fraction genome altered data used were derived and available in updated form the cBio Portal (RRID: SCR_002877, http://www.cbioportal.org/study?id=msk_impact_2017, samples and annotation used at time of analysis available as *Supplementary file 7* and *8*). A cohort of 7305 primary tumor cases across 53 different cancer types and a cohort 5907 metastatic tumor cases, across 47 different cancer types, were identified. Within the primary and metastatic disease cohorts, we excluded patients with unknown tumor CNA burden, overall survival status, unreported follow-up time, death or censoring immediately after treatment, unknown cancer type, and unknown mutation burden. The final cohort used here therefore included 6610 and 4864 patients, respectively. Within both of these cohorts, univariate Cox models were used to determine whether CNA or mutation burden is associated with overall survival. Reported follow-up time was used. As it is probable that the association between tumor CNA burden and survival likely varies based on the particular cancer type, we focused on patients with the ten most prevalent cancer types in both of the respective cohorts (*Supplementary file 2*, 5198 and 3886 patients with primary and metastatic disease respectively) and proceeded with a meta-analysis in order to stratify by cancer type. In particular, we utilized a multivariable Cox model, adjusting for mutation burden for each cancer type and extracted the effect size. The effect size for each cancer type was then entered into a meta-analysis using the *metan* command in Stata. Both fixed effects and random effects estimates were calculated. Fixed effects weights were calculated using inverse-variance weighting, *metan* weights were calculated using the DerSimonian and Laird method.

## Statistical analyses of TCGA cohorts

For analyses of TCGA cohorts, the following published cohorts were filtered for only primary, non-neoadjuvantly treated cases and analyzed: TCGA prostate adenocarcinoma (2015) (*Cancer Genome Atlas Research Network, 2015*), breast carcinoma (*Ciriello et al., 2015*), uterine endometriod cancer (*Cancer Genome Atlas Research Network et al., 2013*), renal clear cell carcinoma (*Cancer Genome Atlas Research Network, 2013*), papillary thyroid carcinoma (*Cancer Genome Atlas Research Network, 2014*), and colorectal

adenocarcinoma (*Cancer Genome Atlas Network, 2012*). The number of cases and exclusions based on unavailable data are detailed in *Supplementary file 9*. Cox regression was used to test the association of tumor CNA burden as a continuous variable with (i) cancer free status and (ii) overall survival in univariate models and in multivariable models with disease stage. For the TCGA colorectal cancer cohort, tumor stage was also used. For the TCGA prostate adenocarcinoma cohort, multivariable Cox regression models that included Gleason score, mutation count, ploidy, and/or ABSOLUTE purity (*Carter et al., 2012*) originally reported with this cohort were also used. Analyses including purity exclude 37 patients without absolute tumor purity measured, resulting in analysis with 243 men, 29 of whom had BCR, and a median followup time for survivors of 20.1 (7.0, 37.9) months.

Data access. The conservative treatment TAPG copy number cohort array data was deposited in NCBI GEO (Gene Expression Omnibus, under accession number GSE103665 (https://www.ncbi.nlm.nih.gov/geo/query/acc.cgi?acc=GSE103665).

# Acknowledgements

We thank the members of the Prostate Cancer Oncogenome Group for critical contributions. This work was supported by HHMI (CLS), CA193837, CA092629, CA155169, the Prostate Cancer Foundation Young Investigator Award (to KKY), Orchid (DMB). We thank the MSKCC Integrated Genomics Operation Core for technical work. The MSKCC Integrated Genomics Operation Core is funded by P30 CA08748, Cycle for Survival and the Marie-Josée and Henry R Kravis Center for Molecular Oncology.

# Additional information

## Competing interests

Charles L Sawyers: Senior Editor, *eLife;* Board of Directors of Novartis; co-founder of ORIC Pharm; co-inventor of enzalutamide and apalutamide; Science advisor to Agios, Beigene, Blueprint, Column Group, Foghorn, Housey Pharma, Nextech, KSQ, Petra and PMV; co-founder of Seragon, purchased by Genentech/Roche in 2014. The other authors declare that no competing interests exist.

## Funding

| Funder | Grant reference number | Author |
| --- | --- | --- |
| Prostate Cancer Foundation | | Kamlesh Yadav |
| American Cancer Society | RSG-15-067-01-TBG | Barry Taylor |
| Prostate Cancer Foundation | | Barry Taylor |
| National Cancer Institute | R01 CA204749 | Barry Taylor |
| Howard Hughes Medical Institute | | Charles L Sawyers |
| National Institutes of Health | CA193837 | Charles L Sawyers |
| National Institutes of Health | CA092629 | Charles L Sawyers |
| National Institutes of Health | CA155169 | Charles L Sawyers |
| National Institutes of Health | CA008748 | Charles L Sawyers |

The funders had no role in study design, data collection and interpretation, or the decision to submit the work for publication.

## Author contributions

Haley Hieronymus, Conceptualization, Data curation, Formal analysis, Supervision, Investigation, Visualization, Methodology, Writing—original draft, Project administration, Writing—review and editing; Rajmohan Murali, Data curation, Methodology, Project administration; Amy Tin, Data curation, Formal analysis, Visualization, Methodology, Writing—review and editing; Kamlesh Yadav,

Investigation, Project administration; Wassim Abida, Data curation; Henrik Moller, Resources, Data curation, Methodology; Daniel Berney, Resources, Data curation; Howard Scher, Conceptualization, Resources, Supervision; Brett Carver, Conceptualization, Writing—review and editing; Peter Scardino, Conceptualization, Funding acquisition; Nikolaus Schultz, Data curation, Writing—review and editing; Barry Taylor, Conceptualization, Investigation, Visualization, Writing—review and editing; Andrew Vickers, Conceptualization, Formal analysis, Writing—review and editing; Jack Cuzick, Data curation, Validation, Investigation, Project administration; Charles L Sawyers, Conceptualization, Resources, Supervision, Funding acquisition, Writing—review and editing

**Author ORCIDs**
Rajmohan Murali 🆔 http://orcid.org/0000-0001-6988-4295
Charles L Sawyers 🆔 https://orcid.org/0000-0003-4955-6475

**Decision letter and Author response**
Decision letter https://doi.org/10.7554/eLife.37294.026
Author response https://doi.org/10.7554/eLife.37294.027

## Additional files

### Supplementary files
• Supplementary file 1. Cohort characteristics.
DOI: https://doi.org/10.7554/eLife.37294.013

• Supplementary file 2. Distribution of cancer types in IMPACT cohorts.
DOI: https://doi.org/10.7554/eLife.37294.014

• Supplementary file 3. Association between overall survival and CNA burden after adjustment for purity in IMPACT prostate and pan-cancer cohorts. Purity was determined by FACETS (*Shen and Seshan, 2016*).
DOI: https://doi.org/10.7554/eLife.37294.015

• Supplementary file 4. TAPG1 conservative treatment primary prostate CNA cohort.
DOI: https://doi.org/10.7554/eLife.37294.016

• Supplementary file 5. MSK-IMPACT primary prostate tumor cohort annotation.
DOI: https://doi.org/10.7554/eLife.37294.017

• Supplementary file 6. MSK-IMPACT metastatic prostate tumor cohort annotation.
DOI: https://doi.org/10.7554/eLife.37294.018

• Supplementary file 7. MSK-IMPACT primary pan-cancer cohort annotation.
DOI: https://doi.org/10.7554/eLife.37294.019

• Supplementary file 8. MSK-IMPACT metastatic pan-cancer cohort annotation.
DOI: https://doi.org/10.7554/eLife.37294.020

• Supplementary file 9. TCGA Cohort statistics: patient exclusion, events, and follow-up.
DOI: https://doi.org/10.7554/eLife.37294.021

• Transparent reporting form
DOI: https://doi.org/10.7554/eLife.37294.022

### Data availability
All data generated or analysed during this study are included in the manuscript and supporting files and reference materials. The conservative treatment TAPG copy number cohort array data was deposited in NCBI GEO under accession number GSE103665 (https://www.ncbi.nlm.nih.gov/geo/query/acc.cgi?acc=GSE103665).

The following dataset was generated:

| Author(s) | Year | Dataset title | Dataset URL | Database, license, and accessibility information |
|---|---|---|---|---|
| Hieronymus H, Taylor BS, Sawyers | 2018 | Copy number alteration burden is a pan-cancer prognostic factor | https://www.ncbi.nlm.nih.gov/geo/query/acc. | Publicly available at the NCBI Gene |

| CL | associated with metastasis and death in conservatively treated prostate cancer: TAPG1 CNA cohort aCGH data | cgi?acc=GSE103665 | Expression Omnibus (accession no: GSE103665). |
|---|---|---|---|

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
