## [Decision Letter]

Thank you for submitting your article "Tumor copy number alteration burden is a pan-cancer prognostic factor associated with recurrence and death" for consideration by *eLife*. Your article has been reviewed by Jeffrey Settleman as the Senior Editor, Michael Green as the Reviewing Editor, and three reviewers. The following individuals involved in review of your submission have agreed to reveal their identity: Cory Abate-Shen (Reviewer #1); Mark A Rubin (Reviewer #3).

The reviewers have discussed the reviews with one another and the Reviewing Editor has drafted this decision to help you prepare a revised submission.

Summary:

The manuscript analyzed the CNA landscape of conservatively treated prostate cancer in a historic biopsy and transurethral resection cohort and found that increased abundance of CNA in tumors is correlated with recurrence and mortality. This finding was further extended across multiple cancer types by analyzing data from TCGA and other cohorts. Given the prevalence of CNA throughout cancer genomes, this study highlights the potential for incorporating overall CNA burden assessment as a molecular prognostic factor in treatment decision making.

The reviewers agreed that the approach is rigorous and addresses important aspects of biomarker evaluation including taking into account clinical and pathology parameters. The reviewers also felt that the manuscript is clear and well-written. However, the reviewers also expressed a concern about the novelty of the results in the light of previously published work, as elaborated below. The reviewers would like to see your response to these concerns in a point-by-point response letter and in a revised manuscript before committing to a decision regarding appropriateness for publication in *eLife*.

Essential revisions:

1) The authors state that they are surprised by the findings that CNVs have such a strong association with clinical progression for prostate and other cancers. Yet, this is known in the prostate cancer research community and has been previously reported on in different eras based on available technology. Therefore, it would be reasonable for the authors to consider their finding in the context of the prior knowledge. For example, comparative genomic hybridization (CGH) technology led to some of the first genome-wide observations regarding copy number alterations and disease state (Cher et al., 1994; Visakorpi et al., 1995). CGH identified somatic copy number alterations in a high percentage (~75%) of localized PCA. Losses were found to be 5 times more common than gains and most often involved 8p (32%), 13q (32%), 6q (22%), 16q (19%), 18q (19%), and 9p (16%). These early genome-wide studies also suggested that the pattern of copy number alterations changes with disease progression. For example, gains of 7, 8q, and X were more often observed in the CRPC state (Visakorpi et al., 1995). SNP arrays also very nicely accessed CNV (Dumur et al., 2003; Lin et al., 2004). These platforms were used by the TCGA studies to perform a comprehensive copy number analysis for localized PCA (Cancer Genome Atlas Research Network. Electronic address and Cancer Genome Atlas Research, 2015). As well as a prior meta-analysis that showed recurrent CNVs were associated with disease progression (Williams et al., 2014). (Note: This paper is referenced in the text but for a more focused point). These are just a selection. There are others.

2) Related to the above comment, if the major point is that these new datasets add substantially more information, it would be nice to show a down-sizing experiment in silico wherein decreasing the coverage means the signal that supports these associations is lost. What is the minimal coverage needed to make these associations? This is important due to prior work that used older technologies to make similar claims.

3) There is recent published work that appears to be very similar. Pearlman et al., (2018) report, "pan-cancer metastasis potential score (panMPS) based on observed CNAs". panMPS predicts metastasis and metastasis-free survival in cohorts of patients with prostate cancer, triple negative breast cancer and lung adenocarcinoma, and overall survival in the Metabric breast cancer cohort and three cohorts from The Cancer Genome Atlas (TCGA), including prostate, breast and lung adenocarcinoma. And Ross-Adams et al., (2015) explored risk in association with CNV and integrating transcriptomic data, albeit on smaller prostate cancer data sets. These studies and work reported in bioRχiv from the ICGC groups on structural variation might also be valuable to include in a discussion of recent relevant prior work in this field.

4) Is the association between CNA burden and survival in prostate cancer explainable by differences in purity? Impure samples will have low apparent CNA burden.

5) The manuscript points out that "aneuploidy" but not "CNA burden" has previously been associated with outcome. In many cases, "aneuploidy" referred not to overall changes in tumor ploidy, e.g. due to genome doubling, but indeed to something similar to CNA burden. "Chromosomal instability" has also been used to describe the same phenomenon and has been associated with prognosis in multiple cancers. For example: Murayama-Hosokawa et al., 2010 (endometrial cancer); Walther et al., 2008; Mouradov et al., 2013; and Berg et al., 2015 (colorectal cancer, and making the important point that the relationship with CIN or CNA burden may reflect absence of MSI); Karlsson et al., 2007 (breast cancer); and Carter et al., 2006 (multiple cancer types).

6) The overall association between CNA burden and survival may be obscuring negative associations between specific CNAs and survival. Have the authors evaluated the association between specific CNAs and survival, after controlling for overall CNA burden?

---

## [Author Response]

Essential revisions:1) The authors state that they are surprised by the findings that CNVs have such a strong association with clinical progression for prostate and other cancers. Yet, this is known in the prostate cancer research community and has been previously reported on in different eras based on available technology. Therefore, it would be reasonable for the authors to consider their finding in the context of the prior knowledge. For example, comparative genomic hybridization (CGH) technology led to some of the first genome-wide observations regarding copy number alterations and disease state (Cher et al., 1994; Visakorpi et al., 1995). CGH identified somatic copy number alterations in a high percentage (~75%) of localized PCA. Losses were found to be 5 times more common than gains and most often involved 8p (32%), 13q (32%), 6q (22%), 16q (19%), 18q (19%), and 9p (16%). These early genome-wide studies also suggested that the pattern of copy number alterations changes with disease progression. For example, gains of 7, 8q, and X were more often observed in the CRPC state (Visakorpi et al., 1995). SNP arrays also very nicely accessed CNV (Dumur et al., 2003; Lin et al., 2004). These platforms were used by the TCGA studies to perform a comprehensive copy number analysis for localized PCA (Cancer Genome Atlas Research Network. Electronic address and Cancer Genome Atlas Research, 2015). As well as a prior meta-analysis that showed recurrent CNVs were associated with disease progression (Williams et al., 2014). (Note: This paper is referenced in the text but for a more focused point). These are just a selection. There are others.

We thank the reviewers for highlighting this and appreciate the well demonstrated connection of specific CNVs and CNV signatures (derived from supervised analyses) to disease state in prostate cancer. We have expanded our consideration of this in the Introduction and Discussion section. As the reviewers note, CNV patterns or clusters have been associated with Gleason 8+ disease (compared to Gleason 6-7, Williams et al., 2014) and recurrent disease (compared to primary, Visakorpi et al., 1995, 2015). Indeed, our group initially found CNA patterns associated with shorter time to biochemical recurrence in prostate cancer (Taylor et al., 2010). Nonetheless, previous studies have primarily looked at the association of individual or sets of CNVs with pathology and disease state, rather than outcomes themselves and time to outcome.

The novelty of the work presented in our study lies in the association of (1) high quality clinical endpoints across cancer types and in a pan-cancer cohort with (2) the overall level or burden of genomic CNA of a tumor, agnostic to the identity of the component CNAs. Moreover, we adjust for prognostic pathological features which have previously been linked to CNVs, showing that association of CNA burden with outcome is independentof the previously published association with pathology features or disease state. We have expanded on this in the discussion to address the reviewers’ comments.

The key novel insight is the following: specific prognostic CNVs are likely to be cancer specific, whereas here we show the prognostic potential of a generalized measure of overall copy number dysregulation, as well as the feasibility of measuring this dysregulation using a current clinical NGS test. We acknowledge that the concept of CNA burden as a prognostic factor could seem rather obvious, at first glance, if viewed as a summation that includes all specific prognostic CNVs. But CNA burden could include both positive and negative prognostic CNVs that counterbalance this. Our work demonstrates that the prognostic utility of overall CNA burden is not lost in the noise of CNAs that are unassociated or negatively associated with poor outcome.

2) Related to the above comment, if the major point is that these new datasets add substantially more information, it would be nice to show a down-sizing experiment in silico wherein decreasing the coverage means the signal that supports these associations is lost. What is the minimal coverage needed to make these associations? This is important due to prior work that used older technologies to make similar claims.

This is an interesting question that raises a number of issues about coverage, resolution, and depth of genome sampling in assaying tumor copy number, both in newer sequencing approaches and older approaches such as array-based methods. The effective coverage of the genome is similar between targeted sequencing by IMPACT (due to the inclusion of SNP probes to specifically address the coverage question) versus older aCGH approaches (used for our conservative treatment cohort) as well as whole exome sequencing. Specifically, the method of calculating burden from the IMPACT targeted sequencing data incorporates SNP probes throughout the genome. We have updated the description of how CNA burden is calculated from IMPACT to clarify this. To illustrate the similarity in breadth of coverage between these technologies, we have added the prostate cohort analyzed by targeted IMPACT assay to Figure 1A. As shown with this addition, the coverage is similar with the newer clinically approved technology.

Rather than running a downsizing experiment, we took advantage of a pan-cancer IMPACT cohort for which whole exome sequencing was also performed (n=1005) and ran an upsizing experiment. The CNA burden determined by targeted IMPACT assay and by whole exome sequencing is highly correlated (rho=0.88, p-value<0.0001, Figure 2—figure supplement 3). We have added this analysis to the Results section and Discussion section. While the suggestion for a further downsizing experiment is interesting, we feel the IMPACT versus WES comparison addresses the main point raised by the referee. Efforts to further downsize from IMPACT to determine the minimal coverage required for prognosis determination is beyond the scope of this initial report and would best benefit from incorporation of other prognostic clinical markers.

3) There is recent published work that appears to be very similar. Pearlman et al., (2018) report, "pan-cancer metastasis potential score (panMPS) based on observed CNAs". panMPS predicts metastasis and metastasis-free survival in cohorts of patients with prostate cancer, triple negative breast cancer and lung adenocarcinoma, and overall survival in the Metabric breast cancer cohort and three cohorts from The Cancer Genome Atlas (TCGA), including prostate, breast and lung adenocarcinoma. And Ross-Adams et al., (2015) explored risk in association with CNV and integrating transcriptomic data, albeit on smaller prostate cancer data sets. These studies and work reported in bioRχiv from the ICGC groups on structural variation might also be valuable to include in a discussion of recent relevant prior work in this field.

We thank the reviewers for bringing our attention to the recent work by Pearlman et al., (2018). We have added a discussion of this study and how it derives a 295-gene frequency-weighted CNA signature derived by supervised analysis of prostate metastases that is prognostic for metastasis and overall survival in prostate and breast cancer. It is interesting to note that CNA burden outperforms their signature in one prostate cohort, while their signature outperforms CNA burden in the other prostate cohort they analyze (one in which PSA and stage are not prognostic). As a set of CNAs selected from prostate cancer, it may have less generalizable prognostic significance beyond the two cancers examined. We expand on this with some of the recent studies showing that CNA frequency, independent of identity, is associated with survival in pediatric AML (Vujkovic et al., 2017) and NSC lung cancer (Jamal-Hanjani et al., 2017). We have also added some discussion of prepublication results from ICGC and others (available on BioRχiv) on somatic CNA in cancer with regards to CNA, prognosis, and mechanism.

4) Is the association between CNA burden and survival in prostate cancer explainable by differences in purity? Impure samples will have low apparent CNA burden.

This is an important question which we have addressed with several additional analyses. First, we looked at the association between CNA burden and outcome in the TCGA prostate cohort after adjusting for purity using the ABSOLUTE purity metric, which was reported with the first publication of this cohort and represents a standard computationally-determined measure of purity. We find that CNA burden is still prognostic for recurrence after adjusting for sample tumor purity in the TCGA cohort (Table 2 and Results section). It is also still prognostic after adjusting for tumor purity along with Gleason and/or mutation burden, but these additional multivariate models are overfitted and we therefore elected not to include them in the revised manuscript. Next, we analyzed the IMPACT cohorts after adjusting for purity. Importantly, purity in these cohorts is more variable since these samples are not subject to the quality control requirements mandated by TCGA (typically >70% tumor content). We found that CNA burden in prostate tumors falls below the significance of association with overall survival when adjusting for purity determined by FACETS. However, purity-adjusted CNA burden is still significant in the pan-cancer primary tumor cohort and approaches significance in the metastatic tumor cohort (Supplementary file 3 and Results section).

This variability may in part be influenced by the smaller hazard ratio of CNA burden determined by IMPACT (sequencing panel) assay versus arrays (TCGA and TAPG/conservative treatment cohort) in these respective cohorts. It is also worth noting that a subset of samples in the IMPACT prostate cohort could not be included in the purity-adjusted analysis due to computational limitations in the purity calculation. Due to this limitation, we saw a decrease in the significance of association between mortality and CNA burden even before adjusting for purity, given with this smaller number of cases (P = 0.007 in the full cohort versus P = 0.02 in the subset for which a purity calculation was available).

In summary, tumor purity is a potentially significant variable that modulates CNA prognostic significance, and it will be important to continue to explore its effect especially in the context of additional molecular prognostic factors. It is interesting that within the IMPACT cohort, purity itself is associated with outcome and may warrant further investigation in such work. We have added a discussion of these considerations to the Discussion section.

5) The manuscript points out that "aneuploidy" but not "CNA burden" has previously been associated with outcome. In many cases, "aneuploidy" referred not to overall changes in tumor ploidy, e.g. due to genome doubling, but indeed to something similar to CNA burden. "Chromosomal instability" has also been used to describe the same phenomenon and has been associated with prognosis in multiple cancers. For example: Murayama-Hosokawa et al., 2010 (endometrial cancer); Walther et al., 2008; Mouradov et al., 2013; and Berg et al., 2015 (colorectal cancer, and making the important point that the relationship with CIN or CNA burden may reflect absence of MSI); Karlsson et al., 2007 (breast cancer); and Carter et al., 2006 (multiple cancer types).

We thank the referees for calling attention to thee related concepts of CNA burden, aneuploidy, and chromosomal instability. Aneuploidy has predominantly been defined as having a non-diploid DNA index, generally assayed by cytometry (e.g. Walther et al., 2008, which we thank the reviewers for directing our attention to, provides a metastudy which uses this metric for aneuploidy; Danielsen et al., 2016). The ploidy metric (determined by CLONET) used in our manuscript to adjust for aneuploidy in our multivariable regression model reflects the same information as DNA index conventionally used to measure aneuploidy. Indeed, ploidy estimates from SNP and sequencing data, such as the one we use, have been successfully benchmarked against FACS and SKY-determined DNA index in human tumors and cell lines (Carter et al., 2012; Prandi et al., 2014). Therefore, we believe our conclusion that the prognostic significance of CNA burden is independent of aneuploidy reflects the traditional measure of aneuploidy in prognostic significance.

However, we understand that aneuploidy has various, often more mechanistic, definitions (e.g., numerical aneuploidy/CIN and structural aneuploidy/GIN), some of which are not captured by ploidy as numerical DNA index. To address the reviewers’ points, we have clarified how the ploidy used in our analysis was determined and its relationship to ploidy and DNA index used in prognostic studies of aneuploidy or chromosomal instability (Results). We have also qualified our conclusion from this finding in the discussion to take into consideration the way aneuploidy was measured. We have also expanded the discussion of how future work will be needed to determine if CNA burden is prognostic independent of cancer type-specific prognostic classifications, including MSI-positive status in colorectal cancer which, as the reviewers note, is largely exclusive with CIN-positive status and may be involved in the prognostic interplay between CNA burden and CIN status (including Berg et al., 2015).

6) The overall association between CNA burden and survival may be obscuring negative associations between specific CNAs and survival. Have the authors evaluated the association between specific CNAs and survival, after controlling for overall CNA burden?

We looked for the association between survival and selected specific CNAs in multivariable regression models with CNA burden in prostate cancer (Table 3 and Materials and methods section). This analysis was focused on the question of whether the specific CNAs we examined accounted for the prognostic significance of CNA burden, and we found that they did not. These specific CNAs did have negative associations with survival (positive association with mortality) for the most part (or no significant association) and their negative associations with survival were not strengthened by the adjustment for CNA burden through its inclusion in the model.

The question about whether there are specific CNAs that have a positive association with survival that might be obscured by CNA burden is interesting and worthy of future exploration. We believe that many individual prognostic CNAs are likely to be cancer type-specific, and current cohorts unfortunately do not have sufficient size to run individual regressions for all CNAs to uncover these in a rigorous fashion.